# Highly Stable Docetaxel-Loaded Nanoparticles Based on Poly(D,L-lactide)-*b*-Poly(ethylene glycol) for Cancer Treatment: Preparation, Characterization, and In Vitro Cytotoxicity Studies

**DOI:** 10.3390/polym15102296

**Published:** 2023-05-13

**Authors:** Ekaterina V. Kuznetsova, Nikita G. Sedush, Yulia A. Puchkova, Sergei V. Aleshin, Evgeny V. Yastremsky, Alexey A. Nazarov, Sergei N. Chvalun

**Affiliations:** 1National Research Center “Kurchatov Institute”, Moscow 123182, Russia; nsedush@gmail.com (N.G.S.); yellow_jk@mail.ru (Y.A.P.); neos.1991@gmail.com (S.V.A.); s-chvalun@yandex.ru (S.N.C.); 2Enikolopov Institute of Synthetic Polymeric Materials, Russian Academy of Sciences, Moscow 117393, Russia; 3Shubnikov Institute of Crystallography, Federal Science Research Center “Crystallography and Photonics”, Russian Academy of Sciences, Moscow 119333, Russia; 4Department of Chemistry, Lomonosov Moscow State University, Moscow 119991, Russia

**Keywords:** block copolymers, emulsion, poly(lactide), poly(ethylene glycol), nanoparticles, aggregation behavior, stability, docetaxel, drug delivery, freeze-drying

## Abstract

Stability and narrow size distribution are among the main requirements that apply to drug formulations based on polymeric nanoparticles. In this study, we obtained a series of particles based on biodegradable poly(D,L-lactide)-*b*-poly(ethylene glycol) (P(D,L)LA_n_-*b*-PEG_113_) copolymers with varied hydrophobic P(D,L)LA block length *n* from 50 to 1230 monomer units stabilized by poly(vinyl alcohol) (PVA) by a simple “oil-in-water” emulsion method. We found that nanoparticles of P(D,L)LA_n_-*b*-PEG_113_ copolymers with relatively short P(D,L)LA block (*n* ≤ 180) are prone to aggregate in water. P(D,L)LA_n_-*b*-PEG_113_ copolymers with *n* ≥ 680 can form spherical unimodal particles with values of hydrodynamic diameter less than 250 nm and polydispersity less than 0.2. The aggregation behavior of P(D,L)LA_n_-*b*-PEG_113_ particles was elucidated in terms of tethering density and conformation of PEG chains at the P(D,L)LA core. Docetaxel (DTX) loaded nanoparticles based on P(D,L)LA_680_-*b*-PEG_113_ and P(D,L)LA_1230_-*b*-PEG_113_ copolymers were formulated and studied. It was observed that DTX-loaded P(D,L)LA_n_-*b*-PEG_113_ (*n* = 680, 1230) particles are characterized by high thermodynamic and kinetic stability in aqueous medium. The cumulative release of DTX from the P(D,L)LA_n_-*b*-PEG_113_ (*n* = 680, 1230) particles is sustained. An increase in P(D,L)LA block length results in a decrease in DTX release rate. The in vitro antiproliferative activity and selectivity studies revealed that DTX-loaded P(D,L)LA_1230_-*b*-PEG_113_ nanoparticles demonstrate better anticancer performance than free DTX. Favorable freeze-drying conditions for DTX nanoformulation based on P(D,L)LA_1230_-*b*-PEG_113_ particles were also established.

## 1. Introduction

In recent decades, design and development of various nanosized drug delivery systems (DDSs) based on inorganic nanoparticles [1], liposomes [2,3,4], polymeric nanoparticles (PNPs) [5,6,7], dendrimers [8], etc., for treatment of different diseases has gained great interest. The use of nanocarriers can improve the delivery of poorly soluble drugs in water, provide targeted delivery and controlled release of drugs, and increase their stability and bioavailability.

Nanoparticles (NPs) of biodegradable synthetic polymers such as poly(lactide) (PLA) [9,10,11], poly(lactide-*co*-glycolide) (PLGA) [12,13,14], poly(*ε*-caprolactone) (PCL) [15,16], etc., and also natural polymers including gelatin [17], chitosan [18], etc., have been extensively investigated as DDSs due to their biocompatibility and high encapsulation efficacy. The size, shape, and surface charge of PNPs can be tuned in a wide range by changing the molecular composition of the polymer or the NP fabrication technique [19,20,21]. Particles’ cellular uptake, biodistribution, interaction with barriers, circulation time in the bloodstream, etc., can be controlled by adjustable physicochemical properties of PNPs, which provides great opportunities for their biomedical applications [22,23,24].

Another promising group of polymeric DDSs includes NPs based on biodegradable amphiphilic block copolymers that are composed of an inner hydrophobic core and an outer hydrophilic corona. The hydrophobic core of these NPs is usually formed by PLA, PCL, PLGA, polycarbonates, and poly(amino acids) [25]. Biocompatible polymers commonly used for the hydrophilic corona include poly(ethylene glycol) (PEG), poly(N-vinyl pyrrolidone), poly(2-ethyl-2-oxazoline), and phosphocholine-based polymers [25]. A biocompatible outer corona can minimize non-specific interactions of the NPs with the reticuloendothelial system, thereby enabling their prolonged circulation in the bloodstream [26].

PNPs with “core-corona” structure have been studied as vehicles for targeted delivery of various anticancer drugs, e.g., paclitaxel (PTX) [27], docetaxel (DTX) [28], platinum-based drugs [29], as well as for small interfering RNA [30], genes [31], etc. Thus, poly(*ε*-caprolactone)-*b*-poly(ethylene glycol) (PCL_18_-*b*-PEG_45_) NPs loaded with hydrophobic drug PTX showed enhanced cytotoxicity in vitro and improved anti-tumor activity in vitro compared with free PTX on pulmonary carcinoma [27]. Gong et al. showed that entrapping of hydrophobic DTX into N-(tert-butoxycarbonyl)-L-phenylalanine end-capped methoxy poly(ethylene glycol)-*b*-poly(D,L-lactide) (mPEG-*b*-PLA-Phe(Boc)) NPs improved its aqueous solubility by a factor of ~1600 [32]. It was also observed that efficacy of A549 xenograft tumor model inhibition in vitro was higher for DTX-loaded NPs compared with the free drug. Among the advantages of “core-corona” PNPs is their ability to incorporate multiple drugs. Thus, poly(2-methyl-2-oxazoline-*b*-2-butyl-2-oxazoline-*b*-2-methyl-2-oxazoline) (P(MeOx_37_-*b*-BuOx_21_-*b*-MeOx_36_)) NPs loaded with both etoposide and an alkylated cisplatin prodrug for deadly small cell lung cancer treatment were successfully obtained and studied in detail [33].

Poorly soluble in water, DTX is one of the most frequently utilized chemotherapeutic agent for therapy of a wide range of solid tumors such as breast cancer, non-small cell lung cancer, prostate cancer, and gastric cancer. Various polymeric DDSs for DTX have been produced to enhance its pharmacokinetic properties, including NPs based on poly(D,L-lactide)-*b*-poly(ethylene glycol) (P(D,L)LA-*b*-PEG) [34,35], poly(D,L-lactide)-*b*-poly(N-isopropylacrylamide) (P(D,L)LA-*b*-PNIPAM) [36], PCL-*b*-PEG [34,37] and others [38,39,40]. Due to controllable chemical structure and biocompatibility the amphiphilic block copolymers of PEG and P(D,L)LA, PCL or PLGA are still the most promising candidates for DTX delivery. Although numerous studies have been dedicated to the design and development of PNPs for DTX delivery, there are some challenges in their preparation and utilization. One of the key challenges is to produce PNPs that are highly stable in aqueous media. The intravenous nanoformulation of DTX based on the P(D,L)LA_24_-*b*-PEG_45_ copolymer, Nanoxel-PM™, becomes unstable in aqueous medium after 4 h, which indicates its limited stability in vitro [41].

One of the strategies to improve the stability of P(D,L)LA-*b*-PEG NPs loaded with DTX was proposed in ref. [35]. The authors produced drug-loaded P(D,L)LA-*b*-PEG NPs (DTX loading was 5 wt% with respect to the initial copolymer mass) by a thin-film hydration method using both water and weak alkaline solution. It was shown that utilizing phosphate buffer solution (PBS) with pH = 8 provides formation of more stable NPs. Thus, the hydrodynamic diameter value (*D*_h_) of P(D,L)LA-*b*-PEG NPs remained constant (~25 nm) in water and PBS (pH = 8) at 6 and 96 h, respectively. The authors proposed that an increase of the pH value results in higher repulsive forces between NPs and, correspondingly, prevents their agglomeration. In ref. [41], the stability of DTX-loaded P(D,L)LA-*b*-PEG NPs was increased via introduction of charged poly(L-lysine) (PLL) blocks to the copolymer chain. It was observed that PLL_7_-*b*-P(D,L)LA_56_-*b*-PEG_45_ NPs with 5 wt% of DTX (*D*_h_ = 43 ± 2 nm) prepared by the thin-film hydration method remain stable in aqueous media for 24 h. Shi et al. produced DTX-loaded P(D,L)LA_28_-*b*-PEG_45_ NPs stabilized by arginine using the thin-film hydration method and studied their dynamic stability in 5% glucose suspension [42]. It was shown that an increase of the arginine concentration from 2 to 6 mg/mL led to an enhancement of the NPs’ stability: the NPs’ value of *D*_h_ = 28.6 ± 0.5 nm remained unchanged within 2 and 12 h, respectively. The authors suggested that the stabilizing mechanism of NPs is related to the electrostatic interaction between the positively charged guanidinium cation and the negatively charged carboxylate anion as well as the hydrogen bond formation of guanidinium/amino –NH– and carboxylate oxygen [42]. However, it was also noted that an increase of the arginine concentration from 1.5 to 6 mg/mL in dispersion and, consequently, its pH value from 8.77 to 10.23 resulted in undesirable degradation of DTX. In ref. [32], mPEG-*b*-PLA and mPEG-*b*-PLA-Phe(Boc) NPs prepared by the thin-film hydration method were stable within 2 and 24 h at 37 °C, correspondingly. The authors proposed that Boc-L-Phe and DTX can interact through π-π stacking, hydrogen bonding, and hydrophobic forces that resulted in improved stability of DTX-loaded particles.

In the present study, we prepared and characterized nanosized particles based on amphiphilic block copolymers of D,L-lactide and ethylene glycol with various molecular composition as potential vehicles for targeted delivery of hydrophobic anticancer drug DTX. We propose formation of P(D,L)LA_n_-*b*-PEG_113_ (polymerization degree of the P(D,L)LA block *n* varied from 50 to 1230 monomer units) particles stabilized by biocompatible poly(vinyl alcohol) (PVA) by a simple “oil-in-water” method as a promising strategy to improve the stability of DTX nanoformulation. The physicochemical and pharmaceutical properties of PVA-stabilized DTX-loaded P(D,L)LA_n_-*b*-PEG_113_ nanoparticles were characterized by dynamic light scattering for size, thermodynamic and kinetic stability, and the ability to freeze-dry and following reconstitution. Testing also included electrophoretic light scattering for electrokinetic potential, transmission electron microscopy for morphology, high-performance liquid chromatography for drug loading content and its cumulative release rate, and MTT assay for in vitro cytotoxicity and selectivity of DTX loaded into NPs compared with free DTX.

## 2. Materials and Methods

### 2.1. Materials

D,L-lactide (3,6-dimethyl-1,4-dioxane-2,5-dione, 99%,) was purchased from Corbion (Gorinchem, The Netherlands) and recrystallized in butyl acetate (Component Reactive, Moscow, Russia) before use. Methoxy poly(ethylene glycol) (mPEG) with molar mass of 5 kDa was purchased from Sigma-Aldrich (St. Louis, MO, USA) and dried at 150 °C under vacuum for 1 h before use. Stannous (II) 2-ethylhaxanoate (SnOct_2_) (Sigma-Aldrich, St. Louis, MO, USA) and poly(vinyl alcohol) (PVA) (87–90% hydrolyzed) with molar mass of 30–70 kDa (Sigma-Aldrich, St. Louis, MO, USA) were used as received. Docetaxel trihydrate (DTX) (99%) was purchased from Qilu Pharmaceutical Co., Ltd. (Jinan, Shandong, China) and used as received. Cryoprotectants D(-)-mannitol (Honeywell Burdick & Jackson, Seelze, Germany), mPEG with molar mass of 2 kDa (Sigma-Aldrich, St. Louis, MO, USA), and PEG with molar mass of 10 kDa (Sigma-Aldrich, St. Louis, MO, USA) were used as received. All organic solvents (Component Reactive, Moscow, Russia) were of analytical grade and used without further purification. Double distilled water was used for all experiments.

### 2.2. Synthesis of Block Copolymers

Poly(D,L-lactide)-*b*-poly(ethylene glycol) (P(D,L)LA-*b*-PEG) copolymers were synthesized by ring-opening polymerization of D,L-lactide in the presence of mPEG with molar mass of 5 kDa using a previously described procedure with some modifications [29]. The polymerization degree of the P(D,L)LA block (*n*) in P(D,L)LA_n_-*b*-PEG_113_ copolymers was controlled by varying the ratio of D,L-lactide to the hydroxyl group of the mPEG in the reaction mixture. SnOct_2_ (II) (0.06% wt/wt with respect to the amount of D,L-lactide) was used as a catalyst. Polymerization was carried out at 160 °C in a sealed glass ampoule for 30 min to 6 h depending on the P(D,L)LA block length. The synthesized block copolymers were dissolved in tetrahydrofuran (THF) and twice precipitated into cold n-hexane. Finally, the products were dried under vacuum at 100 °C overnight to remove residual solvents.

### 2.3. Characterization of Block Copolymers

Proton nuclear magnetic resonance (^1^H NMR) spectroscopy was performed to determine the chemical composition and number-average molecular weight (*M*_n_) of the synthesized block copolymers. Spectra were recorded on a 300 MHz Bruker WP-250 SY spectrometer (Bruker, Billerica, MA, USA) in 5 mm o.d. sample tubes. For measurements, 30 mg of block copolymer was dissolved in 1 mL of deuterated chloroform (CDCl_3_). The integrals of the peaks corresponding to the P(D,L)LA methine protons (-CH, *δ* = 5.10–5.20 ppm) and PEG methylene protons (-CH_2_˗, *δ* = 3.50–3.80 ppm) were used to calculate the values of *n* of the P(D,L)LA block and *M*_n_ of the copolymer (Appendix A). The integrals of the peaks corresponding to the residual monomer (*δ* = 5.01–5.05 ppm) on the ^1^H NMR spectrum (Appendix A) were less than 1%, confirming its successful removal via precipitation. Thus, the purity of the synthesized copolymers is not less than 99%.

The values of *M*_n_, weight-average molecular weight (*M*_w_), and polydispersity index (PDI) of the synthesized block copolymers were determined by gel permeation chromatography (GPC). Chromatograms were recorded on a Knauer system (Knauer GmbH, Berlin, Germany) consisting of a pump, RI detector, and PLgel 5 μm 10^3^ Å or universal PLgel mixed-C column (Agilent Technologies Inc., Santa Clara, CA, USA) depending on the molecular weight of the polymer (Appendix A). For measurements, 5 mg of block copolymer was dissolved in 1 mL of THF. THF was utilized as a mobile phase with a flow rate of 1 mL/min at 40 °C. GPC column calibration was performed with polystyrene standards (Polymer Laboratories Inc., Essex Road Church Stretton, UK).

Molecular characteristics of the synthesized P(D,L)LA_n_-*b*-PEG_113_ copolymers are presented in Table 1.

### 2.4. Preparation of P(D,L)LA_n_-b-PEG_113_ Nanoparticles

Drug-free nanoparticles (NPs) based on P(D,L)LA_n_-*b*-PEG_113_ copolymers were prepared using a single “oil-in-water” emulsion technique. Briefly, 60 mg of block copolymer was dissolved in 6 mL of dichloromethane (DCM). Then, 12 mL of aqueous solution of PVA stabilizer with the concentration of 5% wt/v was quickly added to the organic phase. The obtained mixture was emulsified in an ice bath for 1 min at 50 W using an ultrasonic homogenizer UP400s (Hielscher Ultrasonic Technology, Teltow, Germany). The organic solvent was evaporated under reduced pressure for 2 h using a rotary evaporator (Heidolph, Schwabach, Germany). Finally, the aqueous suspensions were centrifuged using Optima™ MAX-XP ultracentrifuge (Beckman Coulter, Brea, CA, USA) at 30,000 rpm for 30 min at 23 °C to remove residual organic solvent and excess PVA. The collected NPs were re-dispersed with double distilled water. All samples were washed three times.

To prepare DTX-loaded P(D,L)LA_n_-*b*-PEG_113_ NPs, 1.5 mg of DTX (5% wt/wt with respect to the amount of block copolymer) was preliminary dissolved in DCM. Then, the drug-loaded NPs were prepared similarly to the drug-free P(D,L)LA_n_-*b*-PEG_113_ NPs. To eliminate both free DTX and PVA, obtained NPs were centrifuged, washed three times with double distilled water, and subjected to lyophilization.

### 2.5. Characterization of P(D,L)LA_n_-b-PEG_113_ Nanoparticles

#### 2.5.1. Dynamic Light Scattering (DLS)

The mean particle size and size distribution of the NPs were measured using a Zetasizer Nano ZSP instrument (Malvern Panalytical Ltd., Malvern, UK) equipped with a He-Ne laser with a wavelength of 632.8 nm and a scattering angle of 173°. The aqueous suspensions of the NPs with various concentrations were placed into a plastic cuvette with an optical pathway length of 10 mm. The measurements were carried out at 25 °C. Analysis of autocorrelation functions was performed using Zetasizer software v. 7.11.

#### 2.5.2. Electrophoretic Light Scattering (ELS)

The electrokinetic potential (*ξ*-potential) of the NPs was determined by ELS. The measurements were performed on a Zetasizer Nano ZSP instrument (Malvern Panalytical Ltd., Malvern, UK). The aqueous suspensions of the NPs at concentration of 1 g/L were placed into an U-shaped capillary cuvette. The measurements were carried out at 25 °C.

#### 2.5.3. Transmission Electron Microscopy (TEM)

The morphology of the NPs was observed using a Tecnai™ 12 G2 BioTwin Spirit (FEI Company, Hillsboro, OR, USA) microscope at accelerating voltage of 120 kV with an Eagle 4K detector camera (FEI Company, Hillsboro, OR, USA) operating in the bright field mode. For TEM measurements, the negative staining procedure was used. Thin-carbon-film-coated copper TEM grids were glow-discharged for 15 s with 25 mA plasma current in the Pelco easiGlow system (Ted Pella Inc., Redding, CA, USA). A 3 μL droplet of the aqueous suspension with concentration of 0.5 g/L was deposited to the carbon side of the grid and incubated for 1 min. The carbon side of the grid was rinsed with 10 μL of distilled water, and then 10 μL of uranyl acetate solution with a concentration of 0.5 wt% was applied to the grid and incubated for 30 s. After each step, the excess solution was removed by touching the grid edge with filter paper. Then, the grid was dried for 30 min under ambient conditions.

Based on TEM data, the values of aggregation number (*N*_agg_), core–corona interface area per one tethered PEG chain (*s*_int_), and tethering density of PEG chains on the P(D,L)LA core surface (*σ*) of the P(D,L)LA_n_-*b*-PEG_113_ NPs were estimated. Detailed descriptions of the calculations can be found in the Appendix A.

#### 2.5.4. High-Performance Liquid Chromatography (HPLC)

The content of DTX loaded in the P(D,L)LA_n_-*b*-PEG_113_ NPs was measured by HPLC. Liquid chromatography separations were carried out using an 1200 HPLC system with UV/VIS detector (Agilent, Santa Clara, CA, USA). A reverse-phase gradient separation was achieved on a PerfectSil 300 ODS C18 column (250 × 4.6 mm, 5 µM) (MZ-Analysentechnik GmbH, Mainz, Germany). The column temperature was set at 40 °C and the injection volume was 25 µL. The mobile phases were (A) water and (B) acetonitrile. The flow rate was 1.2 mL/min. Linear gradient steps were used with the initial condition set at 28% B, held for 9 min, increased to 72% B after 38 min, then returned to 28% B at 50 min. The detection wavelength was 232 nm. The retention time of DTX was 27.3 min.

The drug loading content (DLC) and encapsulation efficacy (EE) of DTX-loaded P(D,L)LA_n_-*b*-PEG_113_ NPs were calculated according to the following equations:(1)$$\mathrm{DLC}=\frac{m_{1}^{\mathrm{DTX}}}{m_{\mathrm{NPs}}}\times 100\%$$
(2)$$\mathrm{EE}=\frac{m_{1}^{\mathrm{DTX}}}{m_{0}^{\mathrm{DTX}}}\times 100\%,$$
where $m_{1}^{\mathrm{DTX}}$ is the mass of incorporated DTX in the NPs, $m_{\mathrm{NPs}}$ is the mass of the NPs, and $m_{0}^{\mathrm{DTX}}$ is the initial mass of DTX.

#### 2.5.5. In Vitro Drug Release Profiles

The DTX release behavior of the drug-loaded NPs was evaluated by the dialysis method. Briefly, 12 mL of freshly prepared aqueous suspension of DTX-loaded NPs were transferred to a dialysis membrane tube (MWCO = 3.5 kDa, SnakeSkin^TM^, Thermo Fisher Scientific, Waltham, MA, USA) and then immersed in double distilled water (2 L) at 37 °C under shaking (150 rpm) in the dark. The pH value of fresh double distilled water used for dialysis was 6.7. The pH value was measured at 23 °C using a pH meter S47 SevenMultiTM (Mettler Toledo, Columbus, OH, USA). The aliquots of suspension (1 mL) were withdrawn from the dialysis tube at predetermined intervals and freeze-dried. As the solubility of DTX is extremely low in water, part of the released DTX may be deposited on the tube walls or at its bottom. Thus, to avoid capturing the precipitate, the aliquots were carefully collected from the upper layer of the suspension column. The amount of DTX retained in the freeze-dried NPs was determined following the same HPLC protocol described in the previous section.

#### 2.5.6. Cells and In Vitro Cytotoxicity Assay

The human HCT116 colorectal carcinoma, A549 non-small cell lung carcinoma, MCF7 breast adenocarcinoma, and WI38 non-malignant lung fibroblast cell lines were obtained from the European collection of authenticated cell cultures (ECACC, Salisbury, UK). All cells were grown in a Dulbecco’s modified eagle medium (DMEM) (Gibco™, Paisley, UK) supplemented with 10% fetal bovine serum (Gibco™, São Paulo, Brazil). The cells were cultured in an incubator at 37 °C in a humidified 5% CO_2_ atmosphere and were sub-cultured two times a week. The effect of both blank and DTX-loaded NPs on cell proliferation was evaluated using a common 3-(4,5-dimethylthiazol-2-yl)-2,5-diphenyltetrazolium bromide assay (MTT-assay) [43]. The cells were seeded in 96-well tissue culture plates (TPP, Trasadingen, Switzerland) at a 1 × 10^4^ cells/well in 100 µL of the medium. After overnight incubation at 37 °C, the cells were treated with the solution of tested compounds in DMEM in the concentration range of 0 to 100 µM. DTX was used as a standard. After 72 h of treatment, the solution was removed, a freshly diluted MTT solution (100 µL, 0.5 mg/mL in cell medium) was added to the wells, and the plates were further incubated for 50 min. Subsequently, the medium was removed, and the formazan product was dissolved in 100 μL of dimethyl sulfoxide (Component Reactive, Moscow, Russia). The number of living cells in each well was evaluated by measuring the absorbance at 570 nm using the Zenith 200 rt microplate reader (Biochrom, Cambridge, UK). Each experiment was repeated three times, each concentration was tested in three replicates.

#### 2.5.7. Freeze-Drying of P(D,L)LA_n_-*b*-PEG_113_ Nanoparticles

The aqueous suspensions of DTX-loaded NPs were freeze-dried separately and with different types of lyoprotectants (Table 2). All lyoprotectants were preliminary dissolved in double distilled water, and then 1 mL of the prepared solution with a certain concentration (Table 2) was added to 1 mL of suspension with a concentration of 4 g/L. Then, the samples were frozen at −18 °C. The frozen samples were lyophilized using an Alpha 2–4 LSC system (Martin Christ, Osterode Am Harz, Germany) for 48 h at a pressure of 0.001 mbar and −72 °C. Reconstitution of the freeze-dried NPs was performed by addition of 2 mL of double distilled water under shaking.

### 2.6. Statistical Analysis

All the results are presented as the mean of three independent test runs, and all data are expressed as the mean ± standard deviation.

## 3. Results and Discussions

### 3.1. Characterization of Drug-Free P(D,L)LA_n_-b-PEG_113_ Nanoparticles

Aqueous suspensions of nanoparticles (NPs) based on the synthesized P(D,L)LA_n_-*b*-PEG_113_ copolymers with adjustable hydrophobic P(D,L)LA block length *n* varied from 50 to 1230 monomer units were produced using an “oil-in-water” (O/W) emulsion method in the presence of stabilizing agent PVA and studied with dynamic light scattering (DLS) (Figure 1).

The DLS intensity size distributions curves for all initial suspensions of the P(D,L)LA_n_-*b*-PEG_113_ NPs reveal one peak with a hydrodynamic diameter (*D*_h_) value of more than 100 nm (Figure 1). However, the DLS curves for diluted suspensions with concentration (*C*) of 0.1 g/L of the NPs based on P(D,L)LA_50_-*b*-PEG_113_ and P(D,L)LA_180_-*b*-PEG_113_ copolymers with relatively short P(D,L)LA block length reveal two peaks (Figure 1a and Figure 1b, respectively), which can be attributed to small “core-corona” particles with *D*_h_ values of ~60 nm and their aggregates with *D*_h_ > 100 nm. It is known that the intensity of light scattering of small objects is sufficiently lower than that of large objects [44]. Since peaks corresponding to small particles were detected (Figure 1a,b), one can suppose that the main fraction of individual “core-corona” particles co-exists with the minor fraction of their aggregates. This is qualitatively confirmed by the DLS number size distribution curves (Appendix A). Therefore, aqueous suspensions of P(D,L)LA_680_-*b*-PEG_113_ and P(D,L)LA_1230_-*b*-PEG_113_ particles are characterized by unimodal DLS intensity size distributions over a wide range of the *C* values (Figure 1c and Figure 1d, respectively). Thus, the value of (*D*_h_)_0_ (i.e., the *D*_h_ value at zero concentration of the suspension) for P(D,L)LA_680_-*b*-PEG_113_ and P(D,L)LA_1230_-*b*-PEG_113_ particles was obtained (Table 3). It should be noted that the (*D*_h_)_0_ value for the NPs based on P(D,L)LA_n_-*b*-PEG_113_ copolymers with relatively short P(D,L)LA block length (*n* = 50, 180) could not be estimated, due to bimodality of the DLS intensity size distributions (Figure 1a and Figure 1b, respectively). The *D*_h_ values of both individual P(D,L)LA_n_-*b*-PEG_113_ NPs (*n* = 50, 180) and their clusters, defined as the values corresponding to the first and second peak on the DLS intensity size distribution curves (for suspensions with *C* = 0.1 g/L), are listed in Table 3.

As one can see from Table 3, an increase of *n* value from 680 to 1230 in the P(D,L)LA_n_-*b*-PEG_113_ copolymers resulted in an enhancement of the (*D*_h_)_0_ value of the particles that was in accordance with previous reports [29,45]. The polydispersity index (PDI) of the P(D,L)LA_n_-*b*-PEG_113_ particles prepared by the O/W emulsion method was relatively low, which could be attributed to the stabilizing properties of the PVA during emulsion formation. According to the literature, PVA acts as a surfactant during the emulsification process [46]. The molecules of PVA are adsorbed on the solvent–water interface of the emulsion droplets due to the ability of the hydrophobic part of PVA to bind to the organic phase and that of the PVA hydrophilic part to remain in the aqueous phase, resulting in lower interfacial tension between two phases. After evaporation of the organic solvent under vacuum and elimination of free PVA by repeating washing, the fraction of PVA molecules adsorbed on the core–corona interface of the individual P(D,L)LA_n_-*b*-PEG_113_ NPs acts as steric stabilizer preventing the coalescence of the particles and their subsequent precipitation. The PDI values of particles based on P(D,L)LA_n_-*b*-PEG_113_ with long P(D,L)LA blocks (*n* = 680 and 1230) were less than 0.2 (Table 3), which is considered to be acceptable for drug delivery applications [47].

The negative electrokinetic potential (*ζ*-potential) of the particles (Table 3) can be attributed to the dissociation of P(D,L)LA carboxyl groups on the surface of the particle core. Moreover, the P(D,L)LA block length slightly affects the *ζ*-potential value of the NPs.

The morphology of the polymeric particles was studied by transmission electron microscopy (TEM). All the P(D,L)LA_n_-*b*-PEG_113_ particles were spherical in shape (Figure 2). The values of diameter (*D*) of individual polymeric particles are listed in Table 3. An increase in the P(D,L)LA block length led to higher *D* values of the particles (Table 3). As one can see from Figure 2, TEM images of the particles based on P(D,L)LA_n_-*b*-PEG_113_ copolymers with relatively short P(D,L)LA block lengths (*n* = 50, 180) revealed two populations: individual particles with *D* < 100 nm and their submicron aggregates (Figure 2a,b), which is in accordance with DLS data (Figure 1a,b). It should be noted that the presence in suspension of solvated PEG chains surrounding the P(D,L)LA core led to higher values of NP size evaluated by DLS, compared with those estimated by TEM (Table 3), where the corona chains collapsed during drying [48].

It should be noted that individual polymeric particles based on PLA/PEG amphiphilic block copolymers are prone to secondary aggregation [29,49,50,51]. The secondary aggregation behavior is not completely clear. Yu et al. suggested that the driving force of this process is related to the hydrophobic–hydrophobic interactions between the exposed PLA cores of individual particles due to the weak steric stabilization of PEG chains [49]. However, we suppose that an adsorption of PVA molecules at the core–corona interface of the P(D,L)LA_n_-*b*-PEG_113_ particles during the emulsion process should hinder fusion of the P(D,L)LA cores.

Another possible explanation for secondary aggregate formation is the association of PEG chains [50,52,53]. The clustering of PEG chains in aqueous medium was observed by static and dynamic light scattering, small-angle neutron scattering, and other experimental techniques [54,55,56]. Various reasons of PEG association in water have been proposed, including interchain physical cross-links due to intense hydrogen bonding [57], chain end effects [58], impurities in water [59], etc. The values of an area of the P(D,L)LA core per one tethered PEG chain (*s*_int_) as well as PEG corona tethering density (*σ*) were estimated, making the assumption that the value of *D* evaluated from TEM data was equal to the diameter of the P(D,L)LA core of the particles (a detailed description of the calculation can be found in the Appendix A). As one can see from Table 3, a decrease in hydrophobic P(D,L)LA block length in the range of the P(D,L)LA_n_-*b*-PEG_113_ copolymers resulted in smaller values of *s*_int_ and, correspondingly, larger values of *σ*. It is noteworthy that an increase in density of the particle corona leads to enhanced repulsive interaction between PEG chains and their more stretched conformation. Thus, we suggest that the particles based on P(D,L)LA_50_-*b*-PEG_113_ and P(D,L)LA_180_-*b*-PEG_113_ copolymers with relatively short P(D,L)LA block length exhibited high values of PEG corona tethering density (*σ* ≥ 0.5) and exposure to secondary aggregation because of the association of hydrophilic PEG chains due to their elongated conformation.

While the secondary aggregation of polymeric particles used as vehicles for various drugs and their wide size distribution are unfavorable in terms of drug delivery, particles of P(D,L)LA_680_-*b*-PEG_113_ and P(D,L)LA_1230_-*b*-PEG_113_ copolymers with long P(D,L)LA block length loaded with docetaxel (DTX) were produced by the O/W emulsion technique. The physicochemical characteristics of the DTX-loaded particles, i.e., the size, morphology, stability, and release rate of DTX, as well as their cytotoxicity were studied.

### 3.2. Characterization of Docetaxel-Loaded P(D,L)LA_n_-b-PEG_113_ Nanoparticles

The aqueous suspensions of the P(D,L)LA_n_-*b*-PEG_113_ particles loaded with docetaxel (DTX) were studied by DLS. The DLS intensity size distributions for DTX-loaded P(D,L)LA_680_-*b*-PEG_113_ and P(D,L)LA_1230_-*b*-PEG_113_ were unimodal with well-defined peak positions (Figure 3a,b, correspondingly). The values of (*D*_h_)_0_, PDI, and *ζ*-potential of the DTX-loaded particles are listed in Table 4. The influence of the DTX loading on the (*D*_h_)_0_ values of the P(D,L)LA_n_-*b*-PEG_113_ particles was found to be ambiguous; the values of both PDI and *ζ*-potential remained unchanged within experimental uncertainly. Moreover, the DTX loading did not affect the morphology of the studied particles (Figure 3c,f).

Drug loading content (DLC) in the P(D,L)LA_n_-*b*-PEG_113_ (*n* = 680, 1230) particles as well as encapsulation efficacy (EE) of the DTX were evaluated using high-performance liquid chromatography (HPLC). The values of DLC and EE for DTX-loaded P(D,L)LA_n_-*b*-PEG_113_ particles are listed in Table 4.

As can be seen in Table 4, an increase in P(D,L)LA block length leads to a higher amount of the DTX loaded into the P(D,L)LA_n_-*b*-PEG_113_ particles. It is known that the physical encapsulation of hydrophobic drugs into polymeric particles is mainly driven by the hydrophobic interactions between drug molecules and hydrophobic segments of polymer [60]. The higher DLC of the P(D,L)LA_1230_-*b*-PEG_113_ particles can be attributed to the larger P(D,L)LA domain as well as stronger P(D,L)LA hydrophobicity that favors an encapsulation of hydrophobic DTX into the P(D,L)LA core of the particles.

It is known that polymeric particles for drug delivery should be thermodynamically stable, i.e., stable under dissolution, to prevent their break-up into free polymeric chains and the “burst” release of loaded substances under injection. The aqueous suspensions of DTX-loaded P(D,L)LA_680_-*b*-PEG_113_ and P(D,L)LA_1230_-*b*-PEG_113_ particles with *C* of 4 and 5 g/L, respectively, were diluted 1000 times with double distilled water and then studied by DLS. Figure 3 shows that the DLS intensity size distributions of the particles loaded with DTX remained unimodal with constant peak position under dilution, confirming their thermodynamic stability.

One of the requirements applicable to polymeric particles used as drug delivery systems is their kinetic stability. To study kinetic stability of DTX-loaded P(D,L)LA_680_-*b*-PEG_113_ and P(D,L)LA_1230_-*b*-PEG_113_ particles, aqueous suspensions were stored at both 25 and 37 °C for two months, and the variation in *D*_h_ values of the particles with time was monitored by DLS. It was observed that the size of the P(D,L)LA_n_-*b*-PEG_113_ (*n* = 680, 1230) particles loaded with DTX remained unchanged (within experimental uncertainly) during the experiment (8 weeks) at both temperatures (Figure 4). It should be noted that obtained DTX-loaded P(D,L)LA_n_-*b*-PEG_113_ particles stabilized by PVA showed enhanced kinetic stability compared with other DTX delivery systems based on PLA/PEG copolymers [32,42].

### 3.3. Docetaxel Release Studies

The cumulative release profiles of DTX from the PVA-stabilized P(D,L)LA_680_-*b*-PEG_113_ and P(D,L)LA_1230_-*b*-PEG_113_ particles prepared by the O/W emulsion technique in aqueous solution are shown in Figure 5.

As can be observed, the DTX release behavior from the P(D,L)LA core of the P(D,L)LA_n_-*b*-PEG_113_ (*n* = 680, 1230) particles was characterized by a fast initial release of DTX during the first 16 h, followed by its slow and sustained release even up to 72 h. We suppose that the fast release of DTX may be caused by the diffusion of the DTX molecules that were poorly entrapped in the hydrophobic domain, whereas the slower and continuous release could be attributed to the diffusion of the DTX localized in the P(D,L)LA core of the P(D,L)LA_n_-*b*-PEG_113_ particles [61].

The cumulative release of DTX from the P(D,L)LA_680_-*b*-PEG_113_ particles was higher than that from the P(D,L)LA_1230_-*b*-PEG_113_ at the same time point (Figure 5). After 24 h, 52.3 ± 5.7% and 41.4 ± 5.5% of DTX was released from P(D,L)LA_680_-*b*-PEG_113_ and P(D,L)LA_1230_-*b*-PEG_113_ particles, respectively. The higher rate of DTX release from the P(D,L)LA_680_-*b*-PEG_113_ particles could be attributed to their smaller size and the higher hydrophilicity of the core–corona interface of the particles. We suppose that despite almost the same conformation of tethered PEG chains at the core of the P(D,L)LA_n_-*b*-PEG_113_ particles (*n* = 680, 1230), the slight decrease in the value of *σ* from 0.3 to 0.2 nm^−2^ with increase of *n* from 680 to 1230 monomer units, respectively (Table 3), could result in smaller hydrophilicity of the core–corona interface in the case of the P(D,L)LA_1230_-*b*-PEG_113_ particles compared with the P(D,L)LA_680_-*b*-PEG_113_ particles. Therefore, the higher hydrophilicity of the core–corona interface of the P(D,L)LA_680_-*b*-PEG_113_ particles can lead to their higher release rate of hydrophobic DTX.

As was mentioned in the Introduction section, various approaches have been used to modify PLA-*b*-PEG based polymeric systems in order to enhance their stability for DTX delivery. In general, modified particles provide extended sustained release of DTX compared with unmodified PLA-*b*-PEG particles [32,41]. Tan et al. reported that ~80% and 60% of DTX was released within 48 h from mPEG_45_-*b*-P(D,L)LA_24_ and mPEG_45_-*b*-P(D,L)LA_56_-*b*-PLL_8_, respectively [41]. In ref. [32] the authors observed that after 48 h, 72.8% of DTX was released from mPEG-*b*-P(D,L)LA particles, while less than 50% of DTX was released from mPEG-*b*-P(D,L)LA-Phe(Boc). In the present work, we obtained DTX-loaded P(D,L)LA_n_-*b*-PEG_113_ particles stabilized by PVA with a controllable DTX release rate affected by hydrophobic P(D,L)LA block length. An increase in P(D,L)LA block length resulted in a decrease in DTX release rate. Thus, after 48 h, 67.4 ± 6.9% and 55.2 ± 4.5% of DTX was released from P(D,L)LA_680_-*b*-PEG_113_ and P(D,L)LA_1230_-*b*-PEG_113_ particles, correspondingly.

### 3.4. In Vitro Cytotoxicity Studies

The antiproliferative effect of both free DTX and DTX loaded into the P(D,L)LA_n_-*b*-PEG_113_ particles was studied against the breast cancer MCF7, the colon cancer HCT116, non-small cell lung carcinoma A549, and non-malignant lung fibroblast WI38 cell lines. The values of DTX concentration that caused 50% inhibition of cell growth (*IC*_50_) are listed in Table 5. It was observed that the *IC*_50_ values of unincorporated DTX and DTX loaded into the P(D,L)LA_680_-*b*-PEG_113_ particles remained almost the same (within the experimental uncertainly). It should be noted that similar results were shown previously [32,62]. Therefore, the *IC*_50_ values of DTX loaded in the P(D,L)LA_1230_-*b*-PEG_113_ NPs are higher than those of free DTX for all cell lines, which could be attributed to slower drug release from this DTX nanoformulation compared with DTX-loaded P(D,L)LA_680_-*b*-PEG_113_ particles [34,63].

The cytotoxicity of blank PVA-stabilized P(D,L)LA_n_-*b*-PEG_113_ particles was also studied. The incubation of drug-free NPs with cancer cell lines did not inhibit their growth (data not shown).

The selectivity of DTX loaded in the P(D,L)LA_n_-*b*-PEG_113_ NPs was studied using cell lines originated from malignant A549 and non-malignant WI38 lung tumors. The value of the selectivity coefficient *k* for DTX loaded in the particles increased from 1.4 ± 0.4 to 3.0 ± 0.4 with enhancing of the P(D,L)LA block length, whereas the *k* value for free DTX is about 1.2 ± 0.1 (Table 5).

Thus, we suppose that nanoformulation of DTX based on the P(D,L)LA_1230_-*b*-PEG_113_ particles is a promising candidate for drug delivery due to its highly thermodynamic and kinetic stability, sustained DTX release profile, and enhanced selectivity in vitro. Moreover, the nanoscale size of these particles can provide targeted delivery of the DTX to the tumor in passive manner, due to enhanced permeability and retention effect.

### 3.5. Freeze-Drying of Docetaxel-Loaded P(D,L)LA_n_-b-PEG_113_ Particles

The feasibility of freeze-drying for long-term storage and reconstitution of DTX nanoformulation for injection was studied for DTX-loaded P(D,L)LA_1230_-b-PEG_113_ particles. According to DLS data (not shown), the P(D,L)LA_1230_-b-PEG_113_ NPs were unable to reconstitute after freeze-drying in the absence of any lyoprotectants. Thus, we optimized the freeze-drying of the aqueous suspension of DTX-loaded P(D,L)LA_1230_-b-PEG_113_ particles using biocompatible lyoprotectants such as D(-)-mannitol, PVA with molecular weight of 30–70 kDa (PVA_30–70k_), PEG with molecular weight of 10 kDa (PEG_10k_), and methoxy PEG with molecular weight of 2 and 5 kDa (mPEG_2k_ and mPEG_5k_, respectively).

Aqueous solution of D(-)-mannitol was added to the aqueous suspension of DTX-loaded P(D,L)LA_1230_-*b*-PEG_113_ particles (*C* = 4 g/L) so that the final lyoptotectant:particles ratios were 1:4, 1:1, and 2.5:1. The obtained suspensions were freeze-dried and reconstituted by adding double distilled water. As one can see from Appendix A, the DLS intensity size distributions for reconstituting particles were bimodal regardless of the D(-)-mannitol:particles ratio (the PDI values was higher than 0.2). Thus, low molecular weight D(-)-mannitol proved to be an unsuitable lyoprotectant for freeze-drying of the P(D,L)LA_1230_-*b*-PEG_113_ NPs. We suppose that D(-)-mannitol molecules do not provide sufficient steric barriers to hinder the fusion of the polymeric particles under freeze-drying.

Variation of the *D*_h_ as well as PDI values of DTX-loaded P(D,L)LA_1230_-*b*-PEG_113_ particles with lyoprotectant:particles ratios *m*_PVA30–70k_:*m*_NPs_, *m*_PEG10k_:*m*_NPs_, *m_m_*_PEG5k_:*m*_NPs_, and *m*_mPEG2k_:*m*_NPs_ both before freeze-drying and after reconstitution is shown in Figure 6.

As one can see from Figure 6a, an increase in PVA_30–70k_ content resulted in an enhancement in the *D*_h_ value of DTX-loaded P(D,L)LA_1230_-*b*-PEG_113_ particles before freeze-drying. Based on DLS data, the optimal PVA_30–70k_:particle ratio was found to be 2.5:1. In this case, the *D*_h_ value of the particles remains unchanged before freeze-drying and after reconstitution (*D*_h_ = 310 nm), while the PDI value is less than 0.1 (Figure 6a).

Freeze-drying of DTX-loaded P(D,L)LA_1230_-*b*-PEG_113_ particles was also performed in the presence of PEG with various molecular weights as a lyoprotectant. According to our data, PEG with higher molecular weights such as 5 and 10 kDa are unsuitable for the freeze-drying and reconstitution of the studied particles. The *D*_h_ values of the particles increased approximately 1.8–4.2 times, whereas the PDI value enhanced ~5–16 times after particles’ reconstitution depending on the lyoprotectant:particles ratio (Figure 6b,c). Therefore, PEG with molecular weight of 2 kDa could be used as a lyoprotectant for DTX-loaded P(D,L)LA_1230_-*b*-PEG_113_ particles with an optimal mPEG_2k_:particles ratio of 5:1 (Figure 6d).

## 4. Conclusions

The influence of hydrophobic P(D,L)LA block length on the size, morphology, and aggregation behavior of drug-free P(D,L)LA_n_-*b*-PEG_113_ particles produced using an “oil-in-water” emulsion technique was investigated. It was observed that in aqueous medium, individual spherical nanoparticles with hydrodynamic diameter *D*_h_ < 100 nm based on P(D,L)LA_n_-*b*-PEG_113_ copolymers with relatively short P(D,L)LA blocks (*n* = 50, 180 monomer units) are prone to aggregate with the formation of submicron clusters. Therefore, it was shown that P(D,L)LA_680_-*b*-PEG_113_ and P(D,L)LA_1230_-*b*-PEG_113_ copolymers with long P(D,L)LA blocks were formed close to unimodal spherical nanoparticles (*D*_h_ < 250 nm) with low size polydispersity index (the value of PDI was less than 0.2). We suppose that observed differences in the aggregation behavior of the P(D,L)LA_n_-*b*-PEG_113_ nanoparticles are associated with various tethering density and, correspondingly, conformation (i.e., degree of stretching) of PEG chains on the P(D,L)LA core of the particles. Thus, P(D,L)LA_50_-*b*-PEG_113_ and P(D,L)LA_180_-*b*-PEG_113_ nanoparticles with relatively high density of tethered PEG chains (*σ* = 1.1 and 0.5, respectively) are exposed to aggregation because of the higher stretched conformation of PEG chains and their association with one another in aqueous solution.

For preparation of docetaxel nanoformulation, nanoparticles based on P(D,L)LA_n_-*b*-PEG_113_ copolymers with long P(D,L)LA blocks (*n* = 680, 1230 monomer units) were selected. It was observed that both docetaxel-loaded P(D,L)LA_680_-*b*-PEG_113_ and P(D,L)LA_1230_-*b*-PEG_113_ particles are characterized by high thermodynamic and kinetic stability in aqueous medium. An increase in P(D,L)LA block length resulted in an increase in docetaxel-loading content and a reduction of its release rate at 37 °C. Cytotoxicity and selectivity studies in vitro demonstrated that docetaxel-loaded P(D,L)LA_1230_-*b*-PEG_113_ nanoparticles maintained better anticancer performance than free docetaxel. The optimal conditions for freeze-drying docetaxel nanoformulation based on P(D,L)LA_1230_-*b*-PEG_113_ particles were also evaluated. It was observed that addition of both PVA with molecular weight of 30–70 kDa and mPEG with molecular weight of 2 kDa to the suspension (lyoprotectant:nanoparticles ratios of 2.5:1 and 5:1, respectively) allowed reconstitution of the docetaxel-loaded P(D,L)LA_1230_-*b*-PEG_113_ particles with size and PDI similar to those before freeze-drying. In conclusion, we suggest that docetaxel nanoformulation based on P(D,L)LA_1230_-*b*-PEG_113_ nanoparticles would be a promising candidate for targeted cancer therapy.

## Figures and Tables

**Figure 1 polymers-15-02296-f001:**
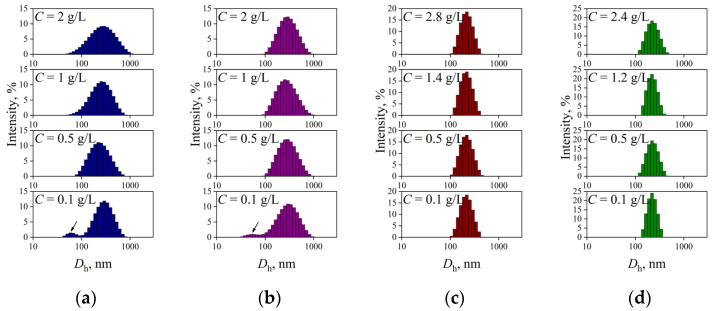
DLS intensity size distribution curves for aqueous suspensions with various concentration (*C*) of PVA-stabilized nanoparticles based on (**a**) P(D,L)LA_50_-*b*-PEG_113_; (**b**) P(D,L)LA_180_-*b*-PEG_113_; (**c**) P(D,L)LA_680_-*b*-PEG_113_; (**d**) P(D,L)LA_1230_-*b*-PEG_113_ copolymers produced using an “oil-in-water” emulsion method. Populations of individual “core-corona” particles of (**a**) P(D,L)LA_50_-*b*-PEG_113_ and (**b**) P(D,L)LA_180_-*b*-PEG_113_ are represented by arrows.

**Figure 2 polymers-15-02296-f002:**
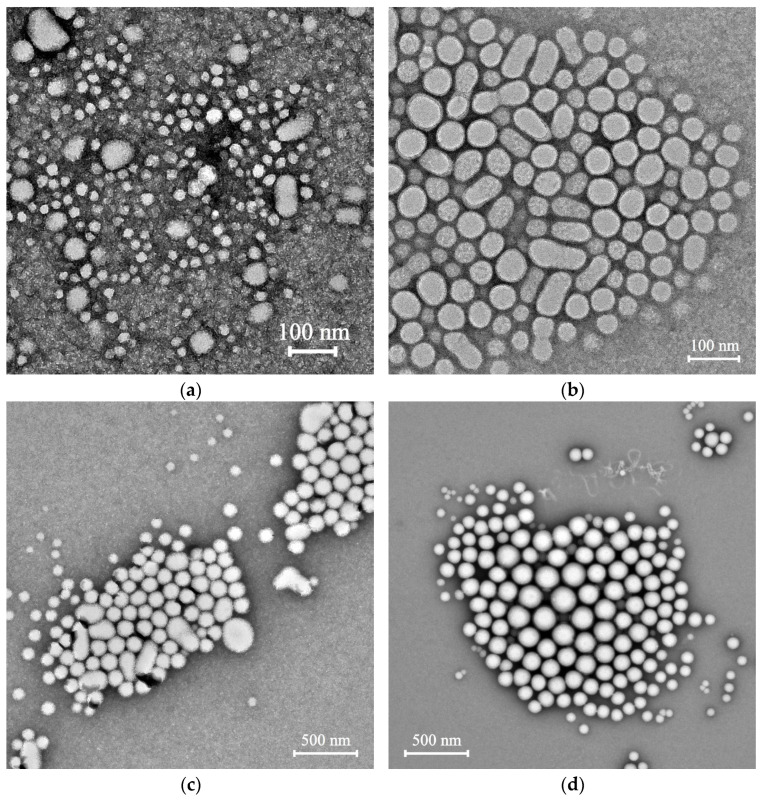
TEM images of produced by an “oil-in-water” emulsion method PVA-stabilized particles based on (**a**) P(D,L)LA_50_-*b*-PEG_113_; (**b**) P(D,L)LA_180_-*b*-PEG_113_; (**c**) P(D,L)LA_680_-*b*-PEG_113_; (**d**) P(D,L)LA_1230_-*b*-PEG_113_ copolymers. The concentration of suspensions was 0.5 g/L.

**Figure 3 polymers-15-02296-f003:**
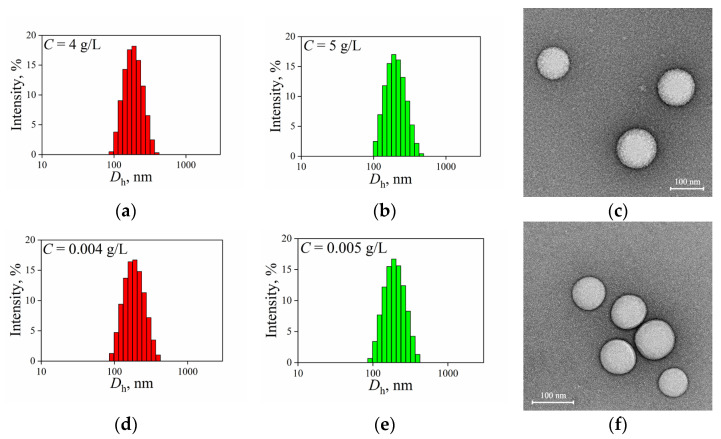
DLS intensity size distribution curves for initial aqueous suspensions of docetaxel-loaded (**a**) P(D,L)LA_680_-*b*-PEG_113_; (**b**) P(D,L)LA_1230_-*b*-PEG_113_ nanoparticles. DLS intensity size distribution curves for diluted 1000 times aqueous suspensions of docetaxel-loaded (**d**) P(D,L)LA_680_-*b*-PEG_113_; (**e**) P(D,L)LA_1230_-*b*-PEG_113_ nanoparticles. Representative TEM images of docetaxel-loaded (**c**) P(D,L)LA_680_-*b*-PEG_113_; (**f**) P(D,L)LA_1230_-*b*-PEG_113_ nanoparticles.

**Figure 4 polymers-15-02296-f004:**
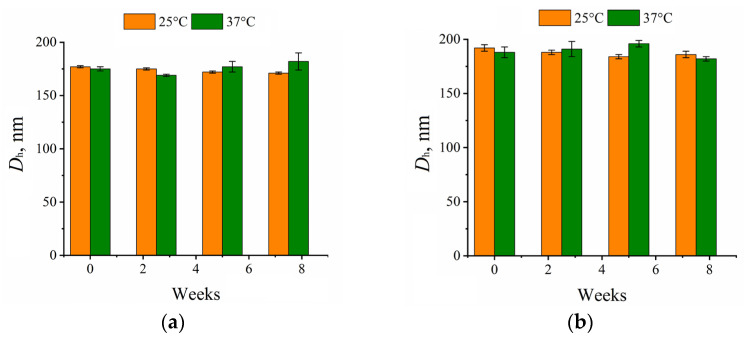
Variation with time of the hydrodynamic diameter average values (*D*_h_, nm) of docetaxel-loaded (**a**) P(D,L)LA_680_-*b*-PEG_113_; (**b**) P(D,L)LA_1230_-*b*-PEG_113_ particles at 25 °C and 37 °C. The concentration of suspensions *C* = 1 g/L.

**Figure 5 polymers-15-02296-f005:**
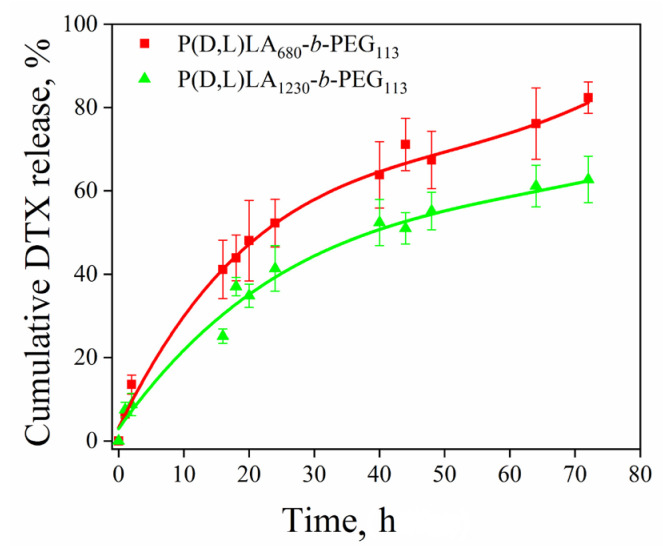
Cumulative release of docetaxel from the P(D,L)LA_n_-*b*-PEG_113_ (*n* = 680, 1230) particles at 37 °C in the dark.

**Figure 6 polymers-15-02296-f006:**
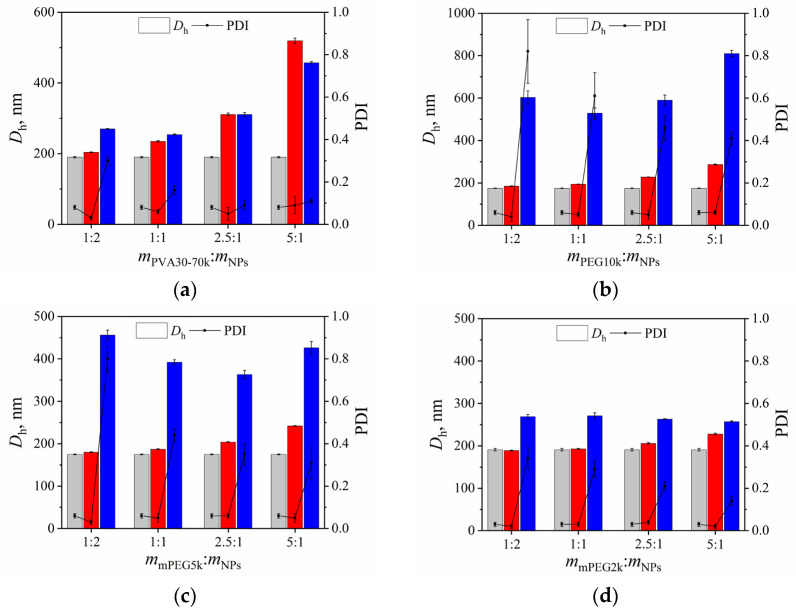
The values of hydrodynamic diameter (*D*_h_, nm) (column) and polydispersity index (PDI) (line) of initial docetaxel-loaded P(D,L)LA_1230_-*b*-PEG_113_ nanoparticles (grey column) after addition of (**a**) PVA_30–70k_; (**b**) PEG_10k_; (**c**) mPEG_5k_; (**d**) mPEG_2k_ as a lyoprotectant before freeze-drying (red column), and after freeze-drying and following reconstitution (blue column).

**Table 1 polymers-15-02296-t001:** Molecular characteristics of the synthesized block copolymers of D,L-lactide and ethylene glycol.

Block Copolymer	*M*_n_ ^1^, kDa	*M*_n_ ^2^, kDa	*M*_w_ ^2^, kDa	PDI ^2^
P(D,L)LA_50_-*b*-PEG_113_	8.6	10.8	13.0	1.2
P(D,L)LA_180_-*b*-PEG_113_	18.1	21.3	27.1	1.3
P(D,L)LA_680_-*b*-PEG_113_	53.8	25.7	54.7	2.1
P(D,L)LA_1230_-*b*-PEG_113_	93.5	41.7	121.5	2.9

^1^ Determined by ^1^H NMR. ^2^ Determined by GPC.

**Table 2 polymers-15-02296-t002:** Types and concentrations of lyoprotectants used for freeze-drying of DTX-loaded P(D,L)LA_n_-*b*-PEG_113_ NPs.

Lyoprotectant	Concentration, g/L
D(-)-mannitol	1, 5, 10
PVA (30–70 kDa)	2.5, 5, 12.5, 25
mPEG (2 kDa)	2.5, 5, 12.5, 25
mPEG (5 kDa)	2.5, 5, 12.5, 25
PEG (10 kDa)	2.5, 5, 12.5, 25

**Table 3 polymers-15-02296-t003:** Physicochemical characteristics of PVA-stabilized P(D,L)LA_n_-*b*-PEG_113_ nanoparticles produced using an “oil-in-water” emulsion method.

Sample	(*D*_h_)_0_ ^1^, nm	PDI ^2^	ζ ^3^, mV	*D* ^4^, nm	*s*_int_ ^5^, nm	*σ* ^6^, nm
P(D,L)LA_50_-*b*-PEG_113_	64 ± 13 *	0.32 ± 0.04	−9 ± 2	33 ± 14	0.9	1.1
	310 ± 130 **					
P(D,L)LA_180_-*b*-PEG_113_	56 ± 14 *	0.26 ± 0.02	−15 ± 4	51 ± 20	2.0	0.5
	320 ± 165 **					
P(D,L)LA_680_-*b*-PEG_113_	161 ± 1	0.11 ± 0.02	−13 ± 3	108 ± 30	3.6	0.3
P(D,L)LA_1230_-*b*-PEG_113_	212 ± 1	0.07 ± 0.02	−12 ± 5	142 ± 40	5.0	0.2

^1^ Hydrodynamic diameter of block copolymer NPs defined as the value corresponding to the zero concentration of suspension. ^2^ The value of polydispersity index. ^3^ The value of electrokinetic potential of block copolymer NPs. ^4^ Diameter of block copolymer NPs evaluated from TEM images. ^5^ The value of core–corona interface area per one tethered PEG chain estimated from TEM data. ^6^ The value of tethering density of PEG chains on the P(D,L)LA core surface estimated from TEM data. * Hydrodynamic diameter of individual “core-corona” P(D,L)LA_n_-*b*-PEG_113_ NPs (*n* = 50, 180) (*C* = 0.1 g/L), defined as the value corresponding to the first peak on the DLS intensity size distribution curve (Figure 1a,b). ** Hydrodynamic diameter of aggregates of P(D,L)LA_n_-*b*-PEG_113_ NPs (*n* = 50, 180) (*C* = 0.1 g/L), defined as the value corresponding to the second peak on the DLS intensity size distribution curve (Figure 1a,b).

**Table 4 polymers-15-02296-t004:** Physicochemical characteristics of docetaxel-loaded P(D,L)LA_n_-*b*-PEG_113_ nanoparticles produced using the “oil-in-water” emulsion method.

Sample	(*D*_h_)_0_ ^1^, nm	PDI ^2^	*ζ* ^3^, mV	*D* ^4^, nm	DLC ^5^, wt%	EE ^6^, %
P(D,L)LA_680_-*b*-PEG_113_	175 ± 1	0.08 ± 0.02	−9 ± 6	118 ± 6	0.5 ± 0.1	10.1 ± 2.0
P(D,L)LA_1230_-*b*-PEG_113_	210 ± 2	0.09 ± 0.03	−15 ± 4	129 ± 24	1.2 ± 0.3	24.3 ± 5.2

^1^ Hydrodynamic diameter of block copolymer NPs, defined as the value corresponding to the zero concentration of suspension. ^2^ The value of polydispersity index. ^3^ The value of electrokinetic potential of block copolymer NPs. ^4^ The value of diameter of block copolymer particles evaluated from TEM images. ^5^ DTX loaded content in the block copolymers NPs evaluated by HPLC. ^6^ The value of encapsulation efficacy of DTX in the block copolymer NPs (initial DLC was 5 wt% with respect to the total mass of the P(D,L)LA_n_-*b*-PEG_113_ copolymer).

**Table 5 polymers-15-02296-t005:** The antiproliferative effect of free docetaxel (DTX) and docetaxel loaded into the P(D,L)LA_n_-*b*-PEG_113_ nanoparticles against various cancer cell lines (MCF7, HCT116, and A549) and non-cancer cell line WI38.

Sample	*IC*_50_, nM				*k* ^1^
	MCF7	HCT116	A549	WI38	
DTX	3.6 ± 1.0	1.5 ± 0.1	3.6 ± 0.1	4.3 ± 0.3	1.2 ± 0.1
P(D,L)LA_680_-*b*-PEG_113_ + DTX	4.6 ± 1.0	1.6 ± 0.3	2.9 ± 0.7	4.3 ± 0.9	1.4 ± 0.4
P(D,L)LA_1230_-*b*-PEG_113_ + DTX	12.2 ± 0.3	5.7 ± 0.1	4.9 ± 0.8	14.5 ± 2.8	3.0 ± 0.4

^1^ The coefficient of selectivity (the ratio of the *IC*_50_ value for WI38 to that for A549).

## Data Availability

The data presented in this study are available on request from the corresponding author.

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
