# Peer review of "Highly Stable Docetaxel-Loaded Nanoparticles Based on Poly(D,L-lactide)-b-Poly(ethylene glycol) for Cancer Treatment: Preparation, Characterization, and In Vitro Cytotoxicity Studies"

_polymers, 2023, doi:10.3390/polym15102296_

Round 1
Reviewer 1 Report
The scientific paper "Highly-Stable Docetaxel-Loaded Nanoparticles Based on poly(D, L-lactide)-B-poly(ethylene Glycol) for Cancer Treatment: Preparation, Characterization, and In Vitro Cytotoxicity Studies" aimed to reports on the obtaining, characterization and in vitro cytotoxicity evaluation of some polymeric microparticles containing the chemotherapeutic drug docetaxel.
While the rationale of this study is strongly supported by past literatures, and seems to possess great potential, the manuscript at its current state holds critical concerns regarding its novelty and significance of its results.
I believe that this manuscript (in the present form) does not meet the criteria to be published in this journal, as I have identified some important flaws:
In the introduction, the authors should highlight the innovation of his experimental protocol, the justification for its realization, prior to the objectives.
As nanomaterials are defined as structures possessing, at minimum, one external dimension measuring 1-100 nm, the current obtained drug delivery systems could not be described as nanomaterials, but rather microparticles.
In the discussion section the results obtained should be compared with those achieved by other researchers and discussions should be significantly detailed. Authors should try to explain the theoretical implication as well as the translational application of their research.
Some other aspects were found in this manuscript:
- all abbreviations should be expanded in the first appearance and should not be repeated in order to decongest the text and facilitate the understanding of the information transmitted;
- the abbreviations CDCL3, DMEM, MTT and others should be explained;
- the information about the substances used for the preparation of the microparticles (catalogue code, manufacturer, city, country) is missing,
- information (manufacturer, city, country) about the companies producing some devices used in the synthesis process and for characterization is missing,
- different fonts were used in the text and in some figures;
- a schematic representation of the study would be appreciated;
- spelling check of the text is required;
- overall revision regarding grammatical errors, style and syntax and general use of English is recommended.
Reviewer 2 Report
Kuznetsova et al. report studies on formulations of docetaxel encapsulated in poly(D, L-lactide)-b-poly(ethylene glycol) nanoparticles, further stabilized with PVA. Such diblock copolymers have been extensively utilized for drug encapsulation and delivery. However, the formulation methods reported here show some novelty. Furthermore, the authors provide a thorough study on the structural characteristics of NPs, their solution properties, drug release and in vitro cytotoxicity. The work is interesting and deserves publication after some issues have been clarified.
1. Title: poly(D, L-lactide)-B-poly(ethylene Glycol) should be written as poly(D, L-lactide)-b-poly(ethylene glycol).
2. Some details on the stabilization mechanism of PVA should be provided.
3. Table 3, first two entries: why a bimodal size distribution is observed? What are the reasons for that observation? I expect that a denser micelle corona would result in better stabilization of the micelles.
4. Has the amount of PVA contained in the micelles been determined? What is the exact location of PVA in the nanoparticles?
5. l. 362-363: association of PEG chains leading to secondary aggregates needs further explanation. Why PEG chains associate in an aqueous medium? Why elongation of PEG chains should lead to association.
6. l. 454-456: I expect that hydrophilicity of PEG chains is the same in all cases. Please clarify.
7. l. 508: something is missing here. The sentence seems incomplete.
8. The authors should provide the SEC traces of the copolymers in the supplementary section.
Reviewer 3 Report
This manuscript describes the preparation of P(D,L)LAn-b-PEG113 with different n values. resulting in particles with PDI less than 0.2 for P(D,L)LA680-b-PEG113 and P(D,L)LA1230-b-PEG113. These two types of nanoparticles were then used for the delivery of docetaxel. these are interesting findings that are appropriate for publication in Polymers, after some minor corrections are done:
1- Regarding Figure 1, the DLS size by volume and/or number would be more indicative of the actual proportion of individual “core-corona” NPs compared to aggregates. The DLS by number might actually show that the number of individual NPs might be higher than the number of aggregates for low concentrations in Figures 1a and 1b.
2- In table 3, why is (Dh)0 smaller for P(D,L)LA180-b-PEG113 than for P(D,L)LA50-b-PEG113?
3- Regarding the docetaxel release studies, what was the pH of the double-distilled water used for dialysis? It would have been best to use a buffer (like PBS) at pH 7.4.
4- Line 478, there seems to be a typo. The sentence “Therefore, the IC50 values of DTX loaded in the P(D,L)LA1230-b-PEG113 NPs is lower than that of free DTX on some cell lines” should read “ Therefore, the IC50 values of DTX loaded in the P(D,L)LA680-b-PEG113 NPs is lower than that of free DTX on A545 cell lines” according to the table 5.
5- Line 507, “accordingly” should be replace with “respectively”, and there is no need of “and” at the end of this sentence.
6- Lines 562-564: unless there is a typo in Table 5, this is not true that “docetaxel-loaded P(D,L)LA1230-b-PEG113 nanoparticles maintained better anticancer performance than free docetaxel”
Round 2
Reviewer 1 Report
The authors have significantly revised the manuscript addressing the concern raised. I consider it could be accepted for publication in this journal.